



# Non-linear response of PM2.5 to changes in NO$_x$ and NH$_3$ emissions in the Po basin (Italy): consequences for air quality plans

Philippe Thunis[1], Alain Clappier[2], Matthias Beekmann[3], Jean Philippe Putaud[1], Cornelis
Cuvelier[4], Jessie Madrazo[5], Alexander de Meij[6]

[1] European Commission, Joint Research Centre, Ispra, Italy

[2] Université de Strasbourg, Laboratoire Image Ville Environnement, Strasbourg, France

[3] Laboratoire Interuniversitaire des Systèmes Atmosphériques, UMR CNRS 7583, Université Paris Est Créteil et
Université de Paris, Institut Pierre Simon Laplace, Créteil, France

[4] Ex European Commission, Joint Research Centre, Ispra, Italy

[5] Signa Terre SA, Geneva, Switzerland

[6] MetClim, Varese, Italy

Corresponding author: Philippe Thunis (philippe.thunis@ec.europea,eu)

**Abstract**: Air pollution is one of the main causes of damages to human health in Europe with an estimate of about
380 000 premature deaths per year in the EU28, as the result of exposure to fine particulate matter (PM2.5) only. In
this work, we focus on one specific region in Europe, the Po basin, a region where chemical regimes are the most
complex, showing important non-linear processes, especially those related to interactions between NO$_x$ and NH$_3$. We
analyse the sensitivities of PM2.5 to NO$_x$ and NH$_3$ emissions by means of a set of EMEP simulations performed with
different levels of emission reductions, from 25% up to a total switch-off of those emissions. Both single and combined
precursor reduction scenarios are applied to determine the most efficient emission reduction strategies and quantify
the interactions between NO$_x$ and NH$_3$ emission reductions. The results confirmed the peculiarity of secondary PM2.5
formation in the Po basin, characterised by contrasting chemical regimes within distances of few (hundreds of)
kilometres, as well as strong non-linear responses to emission reductions during wintertime. One of the striking results
is the increase of the PM2.5 concentration levels when NO$_x$ emission reductions are applied in NO$_x$-rich areas, such
as the surroundings of Bergamo. The increased oxidative capacity of the atmosphere is the cause of the increase of
PM2.5 induced by a reduction in NO$_x$ emission. This process can have contributed to the absence of significant PM2.5
concentration decrease during the COVID-19 lockdowns in many European cities. It is important to account for this
process when designing air quality plans, since it could well lead to transitory increases in PM2.5 at some locations
in winter as NO$_x$ emission reduction measures are gradually implemented. While PM2.5 responses to NO$_x$ and NH$_3$
emission reduction show large variations seasonally and spatially, these responses remain close to linear, i.e.
proportional to the emission reduction levels, at least up to -50% because secondary aerosol formation chemical
regimes are not modified by those relatively moderate ranges.

**Keywords:** urban air pollution, air quality planning, non-linearity, chemical regimes

## 1. Introduction

Air pollution is one of the main causes of damages to human health in Europe with an estimate of about 380 000
premature deaths per year in the EU28, as the result of exposure to fine particulate matter (PM2.5) only (EEA,
2020). Many of the exceedances to the EU limit values occur in urban areas where most of the population is
exposed.



PM2.5 is partly emitted directly (primary particles) and partly formed through photo-chemical reactions that involve
gaseous precursors like $SO_x$, $NO_x$, $NH_3$ and non-methane volatile organic compounds (NMVOC) to form secondary
inorganic and organic aerosol (SIA and SOA). The secondary fraction is often dominating the total concentration of
particulate matter in urban areas as shown by e.g., Beekmann et al. (2015) for the Greater Paris region, De Meij et
al. (2006, 2009) or Larsen et al. (2012) in northern Italy, hence the importance to understand the complex chemical
processes that lead to its formation. In particular, it is key to identify the precursors involved in these reactions in
order to target the right sectors of activity in air quality plans to effectively reduce pollution levels. According to the
EDGAR estimates for 2015 (EDGAR, 2020) for Italy, about 90% of the $NH_3$ is directly emitted in the atmosphere
by the agriculture sector while $SO_x$ precursors are predominantly released by the energy production and use
(industrial) sectors, about 90% (EDGAR estimates for 2015). For $NO_x$, emissions are spread among various sectors,
with transport (50%), industry (40%) and agriculture (4%) being the main ones. The gaseous precursors of
secondary organic aerosols (SOA) include a vast range of high- and low-volatility NMVOCs among which biogenic
terpenes and anthropogenic aromatics. The main sources of aromatics in Italy were in 2012 transport (58%), use of
fuels and solvents (32%), and domestic heating (15%).

Regarding SIA, early works used box models with thermodynamic schemes to address the sensitivity of ammonium
nitrate and sulfate concentrations to gaseous $NH_3$, $NO_x$, and $SO_2$ emissions (Watson et al., 1994; Blanchard and
Hidy, 2003). These models were later on integrated into chemical transport models (CTM), in particular to address
the benefit of additional $NH_3$ emissions reductions in addition to already ongoing $SO_2$ and $NO_x$ emission reductions.
For North America, Makar et al. (2009) simulated with a regional CTM that a 30% reduction of ammonia emissions
would lead to about 1 µg/m$^3$ reduction in PM2.5. For Europe, Bessagnet et al. (2014) simulated the effects of a 30%
$NH_3$ emission reduction in addition to those foreseen by the Gothenburg Protocol for 2030, and found that the G-
ratio defined as the ratio between free ammonia and total nitrate (Ansari and Pandis, 1998) was a good predictor for
the efficiency of $NH_3$ reductions on SIA concentrations. These sensitivities to emission reductions are often
governed by complex chemical mechanisms. A well described phenomenon is the release of free ammonia as a
result of decreased $SO_2$ emissions and sulfate formation, which allows for the formation of additional nitrate, for
example Blanchard and Hidy (2003) and Shah et al. (2018) for wintertime PM2.5 over the Eastern US.  For eastern
China, Fu et al. (2017) and Lachatre et al. (2019) showed both from modelling and satellite observations that this
processes leads to strongly increased ammonia tropospheric columns.  Finally, several works compared CTM
simulations to specific observations. For instance, Pay et al. (2012) showed that the G –ratio was generally
underestimated over Europe, inducing that the Caliope model they used could probably overestimate the efficiency
of $NH_3$ emission reductions.  Petetin et al. (2016) came to a similar conclusion comparing CHIMERE CTM
simulations to observations in the Paris region. They ascribed this underestimation to missing $NH_3$ emissions
especially during warmer periods.

The formation of SOA results from even more complex reactions involving photo-chemical oxidation (as for SIA),
nitration, fragmentation, and oligomerisation of gaseous precursors or secondary products (Kroll and Seinfeld, 2008;
Shrivastava et al., 2017). Models generally use simplified parameterizations to calculate the SOA formation yield
from various classes of parent VOCs (Tsigaridis et al., 2014), and comparison with measurements often show that
SOA sources are still missing in models (Huang et al., 2020).

In this work, we focus on one specific region in Europe, the Po basin. In a companion paper (Clappier et al., 2020)
that analyses PM secondary formation chemical regimes across Europe, the Po basin is clearly identified as a
peculiar area where the chemical regime distributions are the most complex, showing important non-linear processes
(Thunis et al., 2013 and 2015; Carnevale, 2020; Bessagnet, 2014), especially those related to interactions between
$NO_x$ and $NH_3$. The Po basin is also one of the pollution hot spots in Europe where the number of days above the
limit values prescribed by the AAQD for PM10 is yet largely exceeded (EEA, 2020). This situation results from the
high emission density in this region and also from the geographical setting of the area, in border of the Alps and
Apennines mountain ranges that lead to very weak winds in the area, favouring the accumulation of atmospheric
pollutants.

We focus the present analysis on the $NH_3$-$NO_x$ chemical processes and describe their spatial and seasonal
variability, which could help to design more effective mitigation strategies. We start by describing the modelling
set-up and detail the series of simulations required to perform our analysis. Section 3 provides a brief overview of
the modelled base-case concentrations. In Section 4, we analyse the sensitivity of PM2.5 concentrations to $NH_3$ and





NOₓ emissions. Section 5 provides an analysis of the non-linearity in PM2.5 response to these emissions while in Section 6, we discuss the implications of these results in terms of mitigation measures and design of air quality plans. Conclusions are finally proposed.

## 2. Methodology

### 2.1 Modelling set-up

The modelling study is performed with the EMEP/MSC-W air quality model, version rv4_17 (Simpson et al., 2012). The emission input consists of gridded annual national emissions ($SO_2$, $NO$, $NO_2$, $NH_3$, NMVOC, CO and primary PM2.5) at $0.1 \times 0.1$ degrees resolution, based on data reported every year by parties to the Convention on Long Range Transboundary Air Pollution (CLRTAP). These emissions are provided for 10 anthropogenic source-sectors classified by SNAP (Selected Nomenclature for Air Pollution) codes (EMEP, 2003). Meteorological input data are
based on forecasts from the Integrated Forecast System (IFS), a global operational forecasting model from the European Centre for Medium-Range Weather Forecasts (ECMWF). Meteorological fields are retrieved at a $0.1 \times 0.1$ degree longitude latitude resolution and are interpolated to the $50 \times 50 \text{km}^2$ polar-stereographic grid projection (EMEP, 2011).

The modelling domain covers the entire Po basin (Figure 1) with a resolution of 0.1 by 0.1-degree resolution (polar
stereographic projection centred at 60 °N) and includes 20 vertical levels. The initial and background concentrations for ozone are based on Logan (1998) climatology, as described in Simpson et al. (2003). For the other species, background/initial conditions are set within the model using functions based on observations (Simpson et al., 2003 and Fagerli et al., 2004). The simulations cover the entire meteorological year 2015. We will not discuss the validation of the base-case simulation, as this is available in other publications (e.g. Simpson et al., 2012) and
regular status reports by EMEP (https://emep.int/mscw/mscw_publications.html).

In this work, we simulated a series of 24 scenarios in which NOₓ and $NH_3$ emissions were reduced independently or simultaneously by 25, 50, 75 and 100% from the base case reference levels. Emission reductions were applied over the entire Po-basin domain for a complete meteorological year (2015).

### 2.2 Spatial and temporal focus

Results are generally presented in terms of maps but three locations within the domain were selected for a more detailed analysis. The locations are Bergamo (Be) in the northern part of the domain, Mantova (Ma) in the central eastern part of the Po basin and Bologna (Bo) in its southern part. As described in the following sections, we will see that these locations show very different behaviours in terms of response to emission changes.

We also aggregate results into 2 seasons: winter and summer that cover the period from November to February and
from April to September, respectively. These two seasons are characteristic of different chemical regimes as illustrated in the following sections. The process to define the temporal bounds of these two seasons is discussed in Annex 1. The two remaining months (March and October) represent transition periods and are not considered in our analysis.

As we only analyse the processes involving inorganic gas-phase precursors, our focus is on secondary inorganic
PM2.5 although most of the results are expressed in terms of total PM2.5 concentrations. The impact of NOₓ emission reductions on SOA concentration is only briefly discussed in section 5.

### 2.3 Indicators

To describe the interactions between $NH_3$ and NOₓ emissions, we use the relationship proposed by Stein and Alpert (1993) and Thunis and Clappier (2014). This relation expresses the change of concentration resulting from a
reduction of both precursors NOₓ and $NH_3$ simultaneously, as the sum of two single concentration changes and an interaction term, as follows:

$$\Delta C^{\alpha}_{NO_xNH_3} = \Delta C^{\alpha}_{NO_x} + \Delta C^{\alpha}_{NH_3} + \hat{C}^{\alpha}_{NO_xNH_3} \qquad (1)$$





Where $\Delta C$ stands for the PM2.5 concentration change (reference minus scenario) for a percentage α emission reduction (thus the term $\Delta C/\alpha$ is defined positive for a concentration reduction, consecutive to an emission reduction) and $\hat{C}$ for the interaction term. We then scale each of these terms by the emission reduction (α) to generate potentials (P) (Thunis and Clappier, 2014).

$$\frac{\Delta C^{\alpha}_{NO_x NH_3}}{\alpha} = P^{\alpha}_{NO_x NH_3} = P^{\alpha}_{NO_x} + P^{\alpha}_{NH_3} + \hat{P}^{\alpha}_{NO_x NH_3} = \frac{\Delta C^{\alpha}_{NO_x}}{\alpha} + \frac{\Delta C^{\alpha}_{NH_3}}{\alpha} + \frac{\hat{C}^{\alpha}_{NO_x NH_3}}{\alpha} \qquad (2)$$

This division by the factor α is a mean to virtually extrapolate the impact resulting from any percentage emission reduction from 1 to 100%. Potentials facilitate the comparison of concentration changes obtained for different emission reduction levels. Indeed, equal potentials imply a linear relationship between emission reductions and concentration changes. For example, $P^{\alpha} = P^{\beta} \Rightarrow \Delta C^{\alpha} = \frac{\alpha}{\beta}\Delta C^{\beta}$, for α and β, two emission reduction levels.

The overall potential is therefore the sum of two single potentials and one interaction term.

A relation between the potentials of combined emission reductions at two levels of intensity, is obtained by writing (2) for two reduction levels α and β and subtract the two equations. This leads to the following relation:

$$P^{\beta}_{NO_x NH_3} = P^{\alpha}_{NO_x NH_3} + \underbrace{P^{\beta}_{NO_x} - P^{\alpha}_{NO_x}}_{\hat{P}^{\beta-\alpha}_{NOx}} + \underbrace{P^{\beta}_{NH_3} - P^{\alpha}_{NH_3}}_{\hat{P}^{\beta-\alpha}_{NH3}} + \underbrace{\hat{P}^{\beta}_{NO_x NH_3} - \hat{P}^{\alpha}_{NO_x NH_3}}_{\hat{P}^{\beta-\alpha}_{NOxNH3}} \qquad (3)$$

or equivalently via equation 2:

$$P^{\beta}_{NO_x NH_3} = P^{\alpha}_{NO_x} + P^{\alpha}_{NH_3} + \underbrace{\hat{P}^{\alpha}_{NO_x NH_3} + \hat{P}^{\beta-\alpha}_{NOx} + \hat{P}^{\beta-\alpha}_{NH3} + \hat{P}^{\beta-\alpha}_{NOxNH3}}_{non-linear\ terms} \qquad (4)$$

While the two first terms on the right hand side of Eq. 4 represent the single potentials, the remaining right hand side terms quantify the magnitude of non-linearities. $\hat{P}^{\alpha}_{NO_x NH_3}$ quantifies the NO$_x$-NH$_3$ interaction at level α, $\hat{P}^{\beta-\alpha}_{NOx}$ and $\hat{P}^{\beta-\alpha}_{NH3}$ are the single non-linearities associated to NO$_x$ and NH$_3$ emissions, respectively, between levels α and β, whereas $\hat{P}^{\beta-\alpha}_{NOxNH3}$ represents the incremental change of the NO$_x$-NH$_3$ interaction between levels α and β.

Information about non-linearity is important to design air quality plans as it informs on the robustness of a given response, i.e. whether or not this response remains valid over a certain range and type of emission reductions. Because air quality models often provide responses for a limited set of scenarios that are then used as a basis to interpolate/extrapolate the responses to other emission reduction levels, robustness shall always be carefully assessed.

In the next section, we present the baseline results in terms of spatial and temporal variations.

### 3. Baseline concentrations of PM2.5 and gaseous inorganic precursors

Before analysing the impact of emission changes on concentrations, it is worth having a look at the baseline concentration fields. In Figure 1, the yearly averaged PM2.5 concentration fields show a widespread pollution plume covering most of the area, with peak values extending in its central part. The maximum modelled yearly values reach 29 μg/m³, that represent an average between maximum winter values (maximum of 59 μg/m³) and minimum summer values (17 μg/m³).

The seasonal fields of PM2.5 clearly show that high yearly average values mostly result from the winter season contributions when more stable atmospheric conditions lead to stagnant conditions, favouring the accumulation of particulate matter in the area (Pernigotti et al., 2014; Raffaelli et al., 2020). The increased emissions from the





residential sector (heating, especially wood burning) also foster this process (Ricciardelli et al., 2017; Hakimzadeh et al., 2020). The relative contribution of secondary inorganic particles (SIA) ranges between 40 and 50%, regardless of the season and is quite homogeneously distributed spatially over the entire area. Strategies targeting SIA have therefore the potential to abate about half of the total PM2.5 concentration.

As mentioned earlier, the secondary inorganic fraction of PM2.5 results from complex atmospheric processes that involve gaseous precursors (mainly $SO_2$, $NO_x$ and $NH_3$), that can be summarized by the two following chemical pathways:

$$NO_2(g) + OH + M \rightarrow HNO_3(g) + M$$
$$HNO_3(g) + NH_3(g) \leftrightarrow NH_4^+(p) + NO_3^-(p) \tag{5}$$

$$SO_2(g) + OH \rightarrow\rightarrow H_2SO_4(g)$$
$$SO_2(aq) + O3, H2O2\ (aq) \rightarrow\rightarrow H_2SO_4(aq)$$
$$H_2SO_4(g) \rightarrow H_2SO_4(p) \rightarrow\rightarrow 2H^+(p) + SO_4^{2-}(p)$$
$$H_2SO_4(p) + 2NH_3(g) \rightarrow 2NH_4^+(p) + SO_4^{2-}(p) \tag{6}$$

where (g) means "gas phase", (aq) aqueous phase, while (p) means "particulate matter", and the character $\rightarrow\rightarrow$ symbolizes a chemical pathway that summarizes a set of underlying reactions.

The second of these pathways is generally slower than the first one, the $NO_2$ oxidation specific time constant being typically some hours to a day, and that of $SO_2$ one to several days (Seinfeld and Pandis, 2006).

The spatial fields for the seasonal average concentrations of these precursors (Figure 2) reflects their emissions spatial patterns resulting in a $NO_2$-rich area that comprises Milan plus its northern districts (and to a lesser extent, Turin), while $NH_3$ is more abundant in the central part of the Po basin, east of Milan, where intensive agriculture practices take place. Finally, high $SO_2$ concentrations are collocated with the $NO_2$-rich areas nearby Milan but with an additional zone around the harbour city of Genova (along the south coast), reflecting the more important $SO_2$ emissions from the shipping sector there. However, $SO_2$ concentrations are about one order of magnitude below those of $NO_2$. Seasonal variations are well marked for $NO_2$ and $SO_2$ concentrations, not so much in terms of minimum and maximum values but rather in terms of spread with an extended high concentration zone covered during wintertime. In contrast, $NH_3$ concentrations remain very similar in summer and winter, both in terms of values and spatial distribution.

In next sections, we simulate a series of emission reduction scenarios to analyse the response of PM2.5 concentration to single and combined reductions of $NH_3$ and $NO_x$.

## 4. Analysis of the SIA formation chemical regimes

### 4.1 Seasonal trends

From a strategic point of view, it is important to know whether $NO_x$ (mostly emitted by the transport, industry and residential sectors) or $NH_3$ (mostly emitted by agriculture) need to be reduced in priority in order to reach effective results on particulate pollution mitigation. In Clappier et al. (2020), we have observed a great heterogeneity in SIA formation chemical regimes across the Po-basin, different regimes being present in limited geographical areas. Here we intend to look at these regimes in more detail.

To analyse in details the chemical regimes in the Po basin, we compare the two single potentials $P_{NO_x}^{25\%}$, $P_{NH_3}^{25\%}$ obtained for moderate emission reduction of 25% (). Figure 3 provides a spatial overview of the difference between these two potentials ($P_{NO_x}^{25\%} - P_{NH_3}^{25\%}$). This indicator tells whether reductions of $NO_x$ of $NH_3$ will lead to the greatest PM2.5 concentration abatement, i.e. if the regime is rather $NO_x$- or $NH_3$-sensitive, with positive and negative values, respectively.





- During summer time (Figure 3 – left), the entire area is under weak $NO_x$-sensitive conditions with a maximum intensity in its central part, between Bergamo and Mantova.
- During wintertime (Figure 3 – right), the situation is contrasted with a wide and intense $NH_3$-sensitive area that appears around and south-eastwards of Bergamo. This area includes big cities like Milan. Other (not as marked) $NH_3$ sensitive regime zones appear nearby coastal areas. Most strongly $NO_x$-sensitive areas are located in the eastern parts of the domain, north of Bologna and Venice. As expected, $NH_3$-sensitive regimes are collocated with the $NO_2$- and $SO_2$- rich areas (Figure 2), whereas $NO_x$-sensitive regimes coincide with $NH_3$-rich areas. The cases of the three selected cities (Bergamo, Mantova and Bologna) representative of the $NH_3$-sensitive, $NO_x$-sensitive and neutral regimes, respectively, are further analysed below. The chemical regimes deducted from the results of emission reduction scenarios can be compared with the maps of the G-ratio (Figure 4), defined by Ansari and Pandis (1998) as the ratio between free ammonia ($NH_3$ and $NH_4^+$) and total nitrate ($HNO_3 + NO_3^-$) after neutralization of $H_2SO_4$. Values of the G-ratio below 1 indicate a $NH_3$-limited chemical regime, while values above 1 characterize a $HNO_3$-limited chemical regime.

$$G = \frac{NH_3(g) + NH_4^+(p) - 2SO_4^{2-}(p)}{HNO_3(g) + NO_3^-(p)} \tag{7}$$

We first note that that formulation of the G-ratio, in terms of abundance of free total ammonia and total nitrate, differs from that of the $NH_3$ and NOx emission sensitive chemical regime. The following discussion tends to show how both diagnostics can be used in a complementary way. During summer, the G-ratio values well above unity indicate a $HNO_3$ limited chemical regime across the Po basin. This corresponds to a NOx sensitive chemical regime in this region. Moreover, the location of the G-ratio maximum between Bergamo and Modena spatially coincides with the most pronounced $NO_x$ sensitive regime, and to a maximum of $NH_3$ concentrations of about 20 µg/m³ (Figure 2). Indeed, $NO_x$ emission reductions lead to $HNO_3$ concentration reductions that is the limiting factor in $NH_4NO_3$ formation according to the G-ratio. During winter, the G-ratio still shows large values in the region south-east of Bergamo, but contrary to winter, the chemical regime is clearly $NH_3$ sensitive. More generally, G-ratio values remain above unity over the whole Po basin while both $NO_x$ and $NH_3$ sensitive chemical regimes prevail in different areas. Thus, even if free total ammonia is more abundant, PM abatement is more sensitive to $NH_3$ than to $NO_x$ emissions. These differences illustrate the impossibility to directly use the G-ratio for air quality management, an interesting result in itself. We will further discuss this interesting behaviour later when addressing non-linearity in section 4.3.

### 4.2 Impact of the emission reduction strength

In this section, we repeat the analysis of Section 4.1 for yearly average concentrations but looking at the step changes of regimes as we progressively remove emissions from the base case situation. (Figure 5). Chemical regimes are well in place for a 25% level reduction (top left) and are only slightly perturbed from 25 to 50% with a reinforcement of the NOx limited regime (top right). . Despite this slight change in intensity, the regimes keep therefore the same spatial patterns. From 50% onward, chemical regimes tend to attenuate and reverse themselves from 75% to 100% (bottom right).

In other words, locations that are $NH_3$-sensitive for moderate reductions become $NO_x$-sensitive for larger reductions, and vice versa.

### 4.3 A summarized overview: the PM2.5 isopleths

Like isopleth plots that show the variations in the $O_3$ concentrations as a function of $NO_x$ and VOC concentrations (Dodge, 1977), similar plots can be created for PM2.5 concentrations as a function of $NO_x$ and $NH_3$ emissions. Simulation results have indeed often been presented as 2D isopleths of PM2.5 or nitrate as a function of precursor emissions, which allows showing in a comprehensive manner their sensitivity, and also, in a qualitative manner, non-linear effects (for example Watson et al., 1994 over USA; Xing et al., 2018 over the Beijing–Tianjin–Hebei region in China). Figure 6 shows the PM2.5 isopleths obtained through an interpolation among the 25 simulation concentration values (these 25 simulations correspond to the white square symbols in each isopleth diagram) at the three locations Be, Ma, and Bo previously defined. The X and Y-axes represent the strengths of the $NH_3$ and $NO_x$





emission sources, respectively. With this type of graphical representation, it is possible to visualize the response of PM2.5 to a $NO_x$ emission change by moving vertically, the reaction to a $NH_3$ emission change by moving horizontally or the reaction to a combined $NO_x$-$NH_3$ emission change by moving diagonally. The larger the number of isopleths we cross on the path (high gradient), the larger the expected impact from an emission reduction will be. A simple theoretical model to generate and interpret these isopleths is proposed in Annex 2.

From the analysis of the isopleths, we note the following points:

- The diagram areas can be divided into two zones separated by a ridge (dashed line in Figure 6). Above the ridge line, PM2.5 is more sensitive to $NH_3$ while below the ridge line, it is more sensitive to $NO_x$ emission reductions. The orientation of the ridge (tending to vertical or horizontal) informs on the type of chemical regime ($NO_x$ or $NH_3$ sensitive, respectively).
- In general the isopleths show a regular pattern with a progressive decrease of PM2.5 concentration when either the $NO_x$ or $NH_3$ emissions are reduced, with the only exception of Bergamo during wintertime, where $NO_x$ reductions up to 70% lead to a small increase of PM2.5 (Figure 6 - top left), whatever the reduction in $NH_3$ emission is. We discuss this particular feature later in this section.
- The efficiency of a $NO_x$ vs $NH_3$ emission reduction varies across locations. We can compare the efficiency of a given reduction by looking at the horizontal (for $NO_x$) and vertical (for $NH_3$) gradients. To support this comparison, we included in each diagram dashed oblique lines that connect similar PM2.5 concentration values for single $NO_x$ and $NH_3$ emission reductions. The more vertical are these lines, the larger is the $NH_3$ abatement impact compared to the $NO_x$ abatement impact. Conversely, the more horizontal they are, the larger is the $NO_x$ abatement impact compared to the $NH_3$ abatement impact. For moderate emission reductions (up to 50%, top right corner), different behaviours are observed: while in Bergamo PM2.5 is more sensitive to $NH_3$ reductions, it is more sensitive to $NO_x$ reductions in Mantova, and equally sensitive to both precursors in Bologna. This corresponds to the spatial patterns of $NO_x$- and $NH_3$-senstivive regimes depicted in Fig. 3.
- Winter- and summertime isopleths show completely different patterns in Bergamo whereas at the two other locations, they remain similar.
- At Mantova where moderate $NO_x$ reductions (e.g. 50%) are the most efficient among the 3 sites, $NH_3$ emission reductions are more efficient than $NO_x$ emission reductions for larger additional reductions (going for example from 75% to 100%). At Bergamo, $NH_3$ reductions are the most effective for moderate reductions whereas $NO_x$ reductions become more effective for larger reductions, as seen by the isopleths spacing. This confirms the findings of reversed chemical regimes for larger additional emission reductions detailed in the previous section.

The special pattern of Bergamo's PM2.5 isopleths during wintertime needs some additional discussion. The increase of the inorganic fraction of PM2.5 as a response to $NO_x$ reductions during wintertime has already been noted by several authors (e.g. Le et al., 2020; Sheng et al., 2018). It has been related to an increase in the oxidizing capacity of the atmosphere and in particular to increased ozone levels. This is due to the prevailing titration of $O_3$ by NO in wintertime high $NO_x$ conditions and in the absence of photochemical ozone production due to reduced solar radiation (Kleinman et al., 1991).

$$NO(g) + O_3(g) \rightarrow NO_2(g) + O_2(g) \tag{8}$$

Accordingly, for a factor of two $NO_x$ emissions decrease, Figure 8 shows roughly a factor of two increase for ozone, while $NO_2$ decreases by a factor of less than two (1.7). These compensating changes result in a small increase in $NO_3$ radical production, the initial step of the major pathway of wintertime $HNO_3$ and nitrate formation (Kenagy et al., 2018).

$$NO_2(g) + O_3(g) \rightarrow NO_3(g) \tag{9}$$

In this pathway, the $NO_3$ radical formation is followed by combination with $NO_2$ to form $N_2O_5$, a reversible process, and heterogeneous $HNO_3$ formation on wet particle surfaces.





$$NO_3(g) + NO_2(g) \leftrightarrow N_2O_5(g) \tag{10}$$

$$N_2O_5(g) + H_2O(\text{aerosol surface}) \rightarrow 2HNO_3(g) \tag{11}$$

The $NO_3$ radical has three major rapid sinks, reaction with NO, photolysis, and reaction with NMVOC's especially terpenes.

$$NO_3(g) + NO(g) \rightarrow 2NO_2(g) \tag{12}$$

$$NO_3(g) + h\nu \rightarrow NO_2(g) + O(g) \tag{13}$$

$$NO_3(g) + h\nu \rightarrow NO_2(g) + O_2(g) \tag{14}$$

$$NO_3(g) + NMVOC(g) \rightarrow \text{products} \tag{15}$$

Reactions 8 to 15 induce additional dependence of $HNO_3$ formation on $NO_x$ species on top of (5), but which partly cancel out, as they are both involved in formation and sink processes.

SOA is formed through a series of chemical reactions of gaseous precursors (mainly volatile , intermediate volatile or semi-volatile organic compounds [VOCs] with the oxidants $O_3$, OH and nitrate radical ($NO_3$) (Li et al., 2011).

$$(S,I)VOC(g) + \text{oxidants} \rightarrow (S,I)OVOC(g) \leftrightarrow SOA(s) \tag{16}$$

Putting all the arguments together, it follows that wintertime ammonium nitrate formation over Bergamo is most probably controlled by $NO_3$ radical formation (9), and its strong non-linearity with respect to $NO_x$ emissions. The fact that this behaviour is observed in Bergamo and not in Mantova or Bologna is due to the much larger $NO_2$ levels in the Bergamo - Milano area (above 50 µg/m$^3$ during winter, Fig.2). Such large $NO_2$ levels are also simulated locally over the Turin area, and also lead to a slightly $NH_3$ sensitive regime there despite a G-ratio well above unity. Beyond 50% $NO_x$ reduction, $NH_4NO_3$ formation decreases because $NO_2$ decreases more rapidly than ozone increases up to its maximum (at 75% $NO_x$ emission reduction, see Figure 7).

This non-linearity between $NO_x$ emission reductions and $HNO_3$ formation also explains the apparent discrepancies with the analysis of G-ratio indicating stronger sensitivity to $HNO_3$. Simply, sensitivity to $HNO_3$ concentration cannot be extrapolated to sensitivity to $NO_x$ emissions in case of the above shown non-linear behaviour. Total nitrate is less abundant, but $NO_x$ emission reductions below about 50% do not reduce it, and $NH_3$ emission reductions are thus more efficient, even if $NH_3$ is more abundant than $HNO_3$. In this respect, the complementary analysis of emission sensitivity and G-ratio gives interesting clues on possible non- linearity in the $NO_x$ – $HNO_3$ relationship.

At Bergamo during winter, the increase in PM2.5 (+ 1.8 µg/m³) arising from a 50% reduction in $NO_x$ emission also results from an increase in sulfate (+ 0.3 □g/m³) and in SOA (+ 0.6 □g/m³) concentrations. Both sulfate and SOA concentrations are closely related to $O_3$ concentrations (Figure 7). The sulfate increase is comparable in magnitude to the nitrate increase, even if sulfate levels are much smaller that nitrate ones. Figure 8 actually shows a strikingly similar response of sulfate, SOA and ozone to $NO_x$ emission reductions (given that $SO_2$ and NMVOC emissions are held constant). Indeed the prevailing wintertime aqueous production of $H_2SO_4$ requires oxidants and in particular ozone (Le et al., 2020; Sheng et al., 2018). In addition, the formation of SOA in both the gas and particulate phases also requires oxidants (Vahedpour et al., 2011; Huang et al., 2020; Feng et al., 2016; Li et al., 2011; Tsimpidi et al., 2010).

Pinder et al. (2008) also note an oxidant limitation for SIA formation over Eastern United states for the 2000 to 2020 period, but in their simulations, it mainly affects sulfate that increases as a result of $NO_x$ emission reductions while





nitrate decreases. This is due to a more important sulfate to nitrate ratio in eastern US than over the Po basin. Fu et al. (2020) derive from combined measurements and modelling that wintertime nitrate during haze events in the

North China Plain (NCP) are nearly insensitive to 30% $NO_x$ emission reductions, because increased ozone levels increase the $NO_x$ to $HNO_3$ conversion efficiency. Following these authors, this conversion also involves the homogeneous $HNO_3$ formation via the $NO_2 + OH$. This reaction also could play a role in the Po basin, in addition to the heterogeneous pathway. Also Leung et al. (2020) simulate that wintertime nitrate abatement in the NCP is buffered with respect to emission reductions by increased oxidant build-up, but also by sulfate to nitrate conversion

by liberating free $NH_3$ through sulfate concentration reduction, which can then enhance nitrate formation. Womack et al. (2019) find an oxidant limitation of nitrate formation over wintertime Utah (USA), and show that nitrate concentration diminishes when reducing VOC emissions.

## 5. Analysis of non-linearities

Clappier et al. (2020) highlighted the specificities of the Po basin area within Europe. They showed that non-

linearities are important in this region. One peculiarity of the Po basin is a marked difference between the chemical regimes encountered within a confined area, that have implications on the linearity of PM2.5 responses to emission changes. In this section, we analyse in more details these non-linearities.

Information on the interaction term at reduction level $\alpha$=25%: $\hat{P}_{NO_xNH_3}^{25\%}$ [= $P_{NO_xNH_3}^{25\%} - (P_{NO_x}^{25\%} + P_{NH_3}^{25\%})$] (first non-linear term in Eq. 4) is provided both spatially and as a scatter plot in Figure 8. This interaction term is almost

constant (~0.9) over the entire domain, regardless of the chemical regimes (Figure 3) and regardless of the season (only the yearly average is shown in Figure 9, but the maps for both seasons are almost identical). At $\alpha$=25%, the interaction terms therefore represents approximately 10% of the overall concentration change ($P_{NO_xNH_3}^{25\%}$). Single impacts would therefore lead to an overestimation (of about 10%) in PM2.5 reduction if added to extrapolate linearly the impact on yearly averaged PM2.5 concentrations of a combined $NO_x$-$NH_3$ emission reduction at 25%.

Qualitatively, this behaviour is explained by the PM2.5 isopleths determined for 3 different sites of the domain (Figure 6). Their hyperbolic shapes indicate this negative non-linearity in the $NH_3$-$NO_x$ interaction. As discussed in Annex 2, linearity would result in isopleths parallel to the descending diagonal lines.

When emission reductions increase from 25% to 50%, three additional non-linear terms are generated (three last terms in equation 4). Figure 9 and 10 provide an overview of these non-linear terms during wintertime and summer

time, respectively. The top left panel of each figure represents their sum, i.e. the total nonlinearity generated between 25 and 50%. The right column shows the non-linearities associated to $NO_x$ and $NH_3$ while the bottom left panel reports the non-linear interaction between the two precursors.

Overall, non-linearities are more important during wintertime than during summertime. This is true both in absolute and relative (not shown) terms. Non-linearities tend to be the largest in between areas that are characterised by well-

marked $NH_3$- or $NO_x$-sensitive regimes (indicated by the blue and red drawn contours). This can be explained by the fact that when one of the two components ($NO_x$ or $NH_3$) is in large excess (compared to the other one), reductions of this compound have then only little impact, implying that both single and combined reductions only involve one compound and are therefore similar.

During wintertime, the overall non-linearity (Figure 9 top left) is dominated by the single $NO_x$-related non-linearity

($\hat{P}_{NO_x}^{50-25\%}$, Figure 9 top right), a singularity in Europe as the Po basin is the only area where this occurs to this extent (Clappier et al., 2020). In the region of Bergamo, the $NO_x$ non-linearity remains relatively weak despite the peculiar PM2.5 responses to $NO_x$ emission reductions (i.e. an increase of PM2.5 concentrations for NOx emission reduction up to 50%, Figure 7). In this $NH_3$ sensitive region, this behaviour can be explained by the strong oxidant limitation of $HNO_3$ formation outlined above (Figure 8). It is worth mentioning that although atypical, this behaviour is quasi-

linear with responses that remain proportional to the emission reduction strength but with a negative slope. Finally, the non-linear interactions (Figure 9, bottom left) are mostly negative, i.e. the sum of impacts resulting from single reductions exceeds the impact of the combined reduction as already observed for the 25% emission reductions.



In relative terms, the overall wintertime non-linearity terms increase by about 30 % when emission reductions increase from 25 and 50% (Figure 11 top, blue points). Note that these non-linearity terms reach larger values in some places of the modelling domain as highlighted by the data point dispersion.

During summer time, non-linear terms are smaller. The overall non-linearity term (Figure 10 top left) is dominated by the NH$_3$-NO$_x$ non-linear interaction term ($\hat{P}_{NO_xNH_3}^{50-25\%}$, Figure 10 bottom left). In summer, PM2.5 concentration is NO$_x$ sensitive in almost all the domain (Figure 3) and emission reductions do not lead to shifts in the chemical regimes. In this case, the interaction terms become more important. The largest non-linear interaction terms appear 390 in the western and northern part of the Po basin. They are everywhere negative, implying again that the sum of the impacts resulting from single reductions exceeds the impact of the combined reduction.

In relative terms, the mean overall summer time non-linearity term increases by about 10 % when emission reductions increase from 25 and 50% (Figure 11 top, red data points). In contrast to wintertime, non-linearities are approximately constant throughout the domain as highlighted by the very low Root Mean Square (RMS) errors.

Figure 11 shows the increases of the overall non-linear terms for emission reduction increasing from 25 to 50%, from 50 to 75%, and finally from 75 to 100%. Regardless of the emission reduction increments, summer time non-linearities remain small all over the domain with regression slope close to 0.9 and very limited data point dispersions (i.e. low RMSE). Wintertime non-linearities further increase significantly from 50% to 75% reduction levels (regression fit parameter close to 1.21) but tend to stabilize between 75 and 100% reduction level (regression fit 400 parameter close to 1.05). It is interesting to note that potentials in winter increase for all segments (all winter points are above the 1:1 line) indicating that the same percentage reduction (25%) gains progressively more impact when more intense reductions are considered.

### 6. Discussion

When designing air quality plans, it is important to identify the key precursors on which to act in priority to hit a 405 specific air quality target, but also to understand the consequences of these choices for various seasons (temporal variations), locations (spatial variability), emission reduction levels (strength) and strategies (combined or single emission reductions). As the information to make this decision is generally incomplete, assessing the robustness of the available model responses is essential. From the results presented here, a few key points appear.

- 410 The seasonal and spatial variabilities in the response of PM2.5 to the reduction of NO$_x$ and NH$_3$ emissions are extremely large, with different and sometimes opposite responses to emission changes. Yearly averages do not represent the appropriate time window to evaluate the impact of such emission reductions and a focus on wintertime (November to February) seems to be the right option, especially because concentrations are larger during this period of the year.

- 415 The responses of PM2.5 to emission reduction plans that cover the whole area (i.e. uniform emission reductions are applied everywhere in the domain) vary from location to location: opposite responses occur within a few hundreds of kilometres for some reduction levels. In the region of Bergamo, PM2.5 response to NOx emission reductions can be negative, meaning an increase of PM2.5 when reducing the NO$_x$ emissions. It is important to combine NO$_x$ and NH$_3$ emission reductions in winter, or to go for stronger 420 emission reductions to make sure these unwanted effects are limited.

- Despite quite important non-linearities, PM2.5 responses to emission reductions are not chaotic. Indeed, regardless of the emission reduction level, the non-linear terms related to NH$_3$ emission reduction and to NO$_x$-NH$_3$ interactions are quite uniform spatially. This is not the case of NO$_x$ emission reduction, for 425 which care must be taken to ensure that the detailed response of PM2.5 is captured.

- Although they are location dependent, PM2.5 isopleths represent an interesting tool to assess the impact of different NO$_x$ and NH$_3$ emission reductions on PM2.5 concentration. They indeed allow visualising in one single diagram the impact of any type of reductions on concentrations in a single grid cell (or set of grid 430 cells).





## 7. Conclusions

In this work, we analysed the sensitivities of PM2.5 to $NO_x$ and $NH_3$ emissions by means of a set of EMEP simulations performed with different levels of emission reductions, from 25% up to a total switch-off of those emissions. Both single and combined precursor reduction scenarios were applied to determine the most efficient emission reduction strategies and quantify the interactions between $NO_x$ and $NH_3$ emission reductions. The results confirmed the peculiarity of secondary inorganic PM2.5 formation in the Po basin suggested by Clappier et al. (2020), characterised by contrasting chemical regimes within distances of few (hundreds of) kilometres, as well as strong non-linear responses to emission reductions during wintertime. One of the striking results is the increase of the PM2.5 concentration levels when $NO_x$ emission reductions are applied in $NO_x$-rich areas, such as the surroundings of Bergamo. The isopleths in the graphs showing PM2.5, nitrate, sulfate, SOA and $O_3$ concentrations as a function of $NH_3$ and $NO_x$ emissions (Figure 7) indicate that the increased oxidative capacity of the atmosphere is the cause of the increase of PM2.5 induced by a reduction in $NO_x$ emission up to -50%. This process can have contributed to the absence of significant PM2.5 concentration decrease during the COVID-19 lockdowns in many European cities (EEA, 2020; Putaud et al., 2020). It is important to account for this process when designing air quality plans, since it could well lead to transitionary increases in PM2.5 at some locations in winter as $NO_x$ emission reduction measures are gradually implemented. At this type of location, mitigation measures that would be optimal in the long-term might not be efficient in the short-term.

Joint analysis of the G-ratio and emissions sensitivities can give a clue if a $NH_3$ sensitive chemical regime is due to either a limitation of $NH_3$ or to a lack of sensitivity of NOx emission reduction, due to a non-linear and opposite behaviour of the $NO_x$ – $HNO_3$ relationship. In this latter case, the chemical regime is $NH_3$ sensitive in terms of $NH_3$ and NOx emission reductions, but $HNO_3$ limited in terms of G-ratio, as observed for the Bergamo – Milano region. Inversely, a positive G-ratio indicating $HNO_3$ limitation does not necessarily indicate a larger sensitivity to NOx than to $NH_3$ emissions. Thus, the results indicate the impossibility to directly use the G-ratio for air quality management, an interesting result in itself. Whereas PM2.5 responses expressed as potentials (i.e. responses projected linearly to a 100% emission reductions) of $NO_x$ and $NH_3$ emission reduction show large variations seasonally and spatially, they are quite insensitive to the emission reduction levels at least up to -50% because the secondary aerosol formation chemical regimes are not modified by those relatively moderate ranges. The response of PM2.5 concentrations to emission reductions is found to be non-linear in certain areas of the Po basin, especially in winter. However, non-linear terms remain quite constant spatially (although they depend on the emission reduction strength), which suggests that potential approaches to model or parameterize these non-linearities might not be too complex to develop. These parametrisations might be useful to interpolate existing modelling responses to situations that are not explicitly modelled, or for their implementation in source – receptor models.

Since sulfate concentrations are low in the Po basin, the impact of $SO_2$ emission reductions was not evaluated here. However, the simulations performed in Clappier et al. (2020) indicate that air quality improvement plans addressing $SO_2$ emissions may still lead to additional PM2.5 decreases. Further works should also test if NMVOC emission would further affect the concentration of oxidants, and subsequently of nitrate (and sulfate) during winter. This depends on the fraction of ozone formed photochemically in the Po basin, compared to the one transported from outside by advection or entrainment.

Finally, it would be important to compare the results obtained in this work from the EMEPrv4_17 model with similar results obtained from other models. With its complex setting, the Po basin represents a good test case for such inter-comparisons.

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





**Annex 1: Selection of seasons**

Chemical regimes show a great variability with time. To select the extension of the seasons, we analyse the monthly behaviour of the $NH_3$ and $NO_x$ responses as well as the interaction terms (Figure 12). The analysis is performed for two locations: Bergamo which lies in a $NH_3$ sensitive zone and Mantova which lies within a $NO_x$ sensitive zone. On these plots, we identify two major seasons with a consistent behaviour: winter from November to February and summer, from April to September. These two seasons are similar at both locations. The two remaining months, April and October are transition months and are not considered in the analysis. It is interesting to note that these two transition periods correspond more or less to the switching time between the $NO_x$ and $NH_3$ concentration time profiles (Figure 12 - right panel). On the latter figures, the $SO_2$ and $NO_2$ temporal evolutions are almost identical, in contrast to $NH_3$.

**Annex 2: A theoretical example for the isopleths**

To facilitate the interpretation of the isopleths diagrams, we use a simple theoretical example that mimics the complex reactions process schematized by equations (5) and (6) above. Our simplified process is described by the following relation: $C_C(x, y) = \min(E_A[x], E_B[y])$ where $C_c$ is the concentration of a compound "c" that is given by the minimum between two emitted species A and B. The concentration depends on the strengths of these emissions, specified by the parameters x and y. For each emission strength (x or y), the two emission species are proportional to their full-scale value (100%): $E_A(x) = xE_A[100]$ and $E_B(y) = yE_B[100]$, respectively. If we choose $E_A(100) \gg E_B(100)$, we create a B-sensitive environment (Figure 13 - left column) and inversely (Figure 13 - right column). If we select mixed situations, representing for example an average of days, during which we alternate between A-sensitive and B-sensitive regimes, we obtain the two bottom isopleth diagrams that represent cases where a larger number of A-sensitive (right) or B-sensitive (left) events are recorded. Although extremely simple, these diagrams illustrate properties that are observed on the real test cases.

Let's take the example of a A-sensitive regime. Similar observations can be made in the case of a B-sensitive regime. We note the following points:

- The diagram area can roughly be divided into two zones separated by a ridge: a top-left triangle where the sensitivity to emission reductions of species "A" dominate and a bottom-right triangle where the sensitivity to B dominates. The slope of the ridge (larger or less than one) informs on the type of regime.
- In the case of a single A-sensitive day (top right) with $E_A(100)=2E_B(100)$, the concentration of compound C remains unchanged for emission reductions of A up to 50% while its concentration react in a linear way to emission changes of B from 0 to 100%. As concentration must drop to zero for full reductions of either A or B, a lower gradient for B than A for small emission reductions implies a reversed behaviour (larger gradient for B than A) for larger reductions. In our simple example, the gradient of B is zero from 0 to 50% but is twice as large as A from 0 to 50% emission strength.
- While the combination of several events (e.g. days) characterised by different regimes leads to smoother isopleths (bottom), the same characteristics can be noted. In particular, the inclination (tending to the horizontal or vertical) provides information on the type of chemical regime.



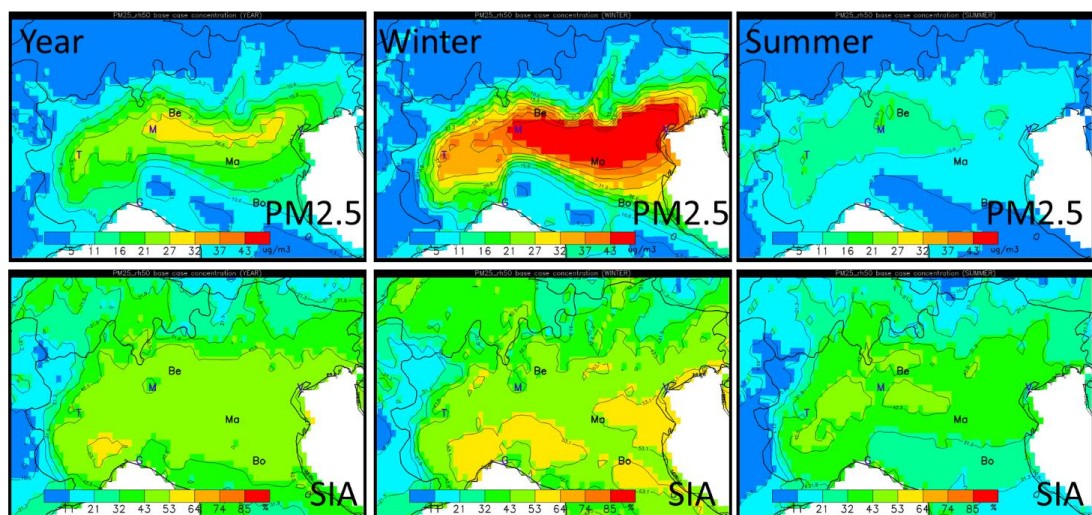

**Figure 1: Seasonal variations of the base case PM2.5 (top row) and SIA (bottom row). The symbols: Be, Ma and Bo represent the locations selected for more detailed analysis: Bergamo (Be), Mantova (Ma) and Bologna (Bo). Other cities are indicated by their first letter for convenience: Venice, Milan, Turin and Genova.**

670

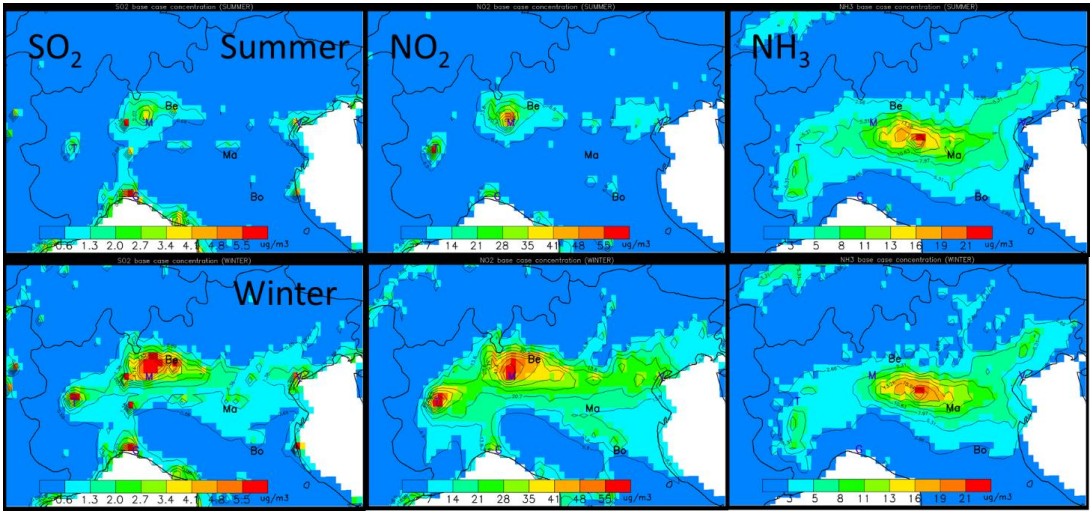

**Figure 2: Summer (top) and winter (bottom) concentration base-case fields for SO2 (left column), NO2 (central column) and NH3 (right column). The symbols: Be, Ma and Bo represent the locations selected for more detailed analysis: Bergamo (Be), Mantova (Ma) and Bologna (Bo). Other cities are indicated by their first letter for convenience: Venice, Milan, Turin and Genova.**

675





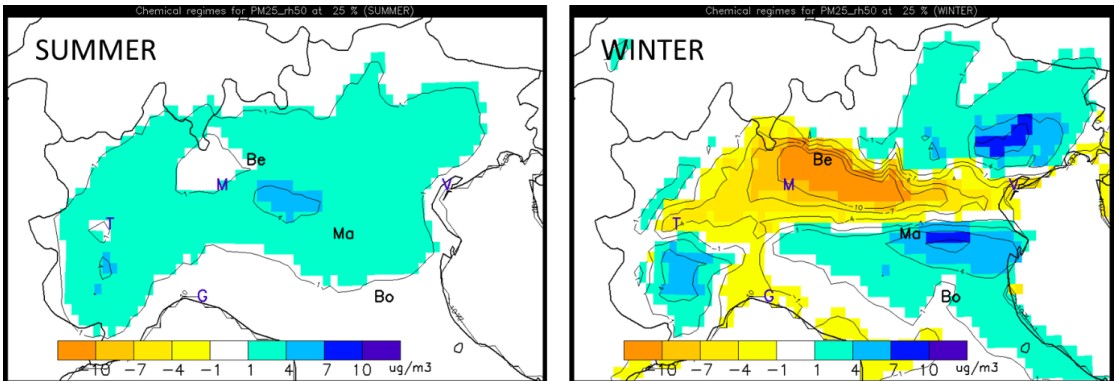

**Figure 3: Winter (right) and Summer (left) chemical regimes obtained at a reduction level of α=25%. The maps represent the $P_{NO_x}^{25\%} - P_{NH_3}^{25\%}$ (unit: µg/m3) indicator that shows the NOx- and NH3-sensitive areas in blue and yellow, respectively. The symbols Be, Ma and Bo indicate the location of Bergamo, Mantova and Bologna, respectively. Other cities are indicated by their first letter for convenience: Venice, Milan, Turin and Genova.**

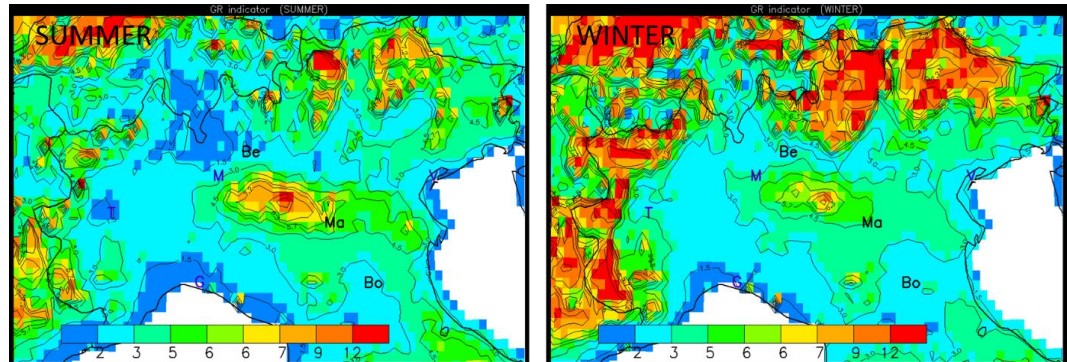

**Figure 4: G-ratio for winter (right) and summer (left) times. The symbols Be, Ma and Bo indicate the location of Bergamo, Mantova and Bologna, respectively. Other cities are indicated by their first letter for convenience: Venice, Milan, Turin and Genova.**



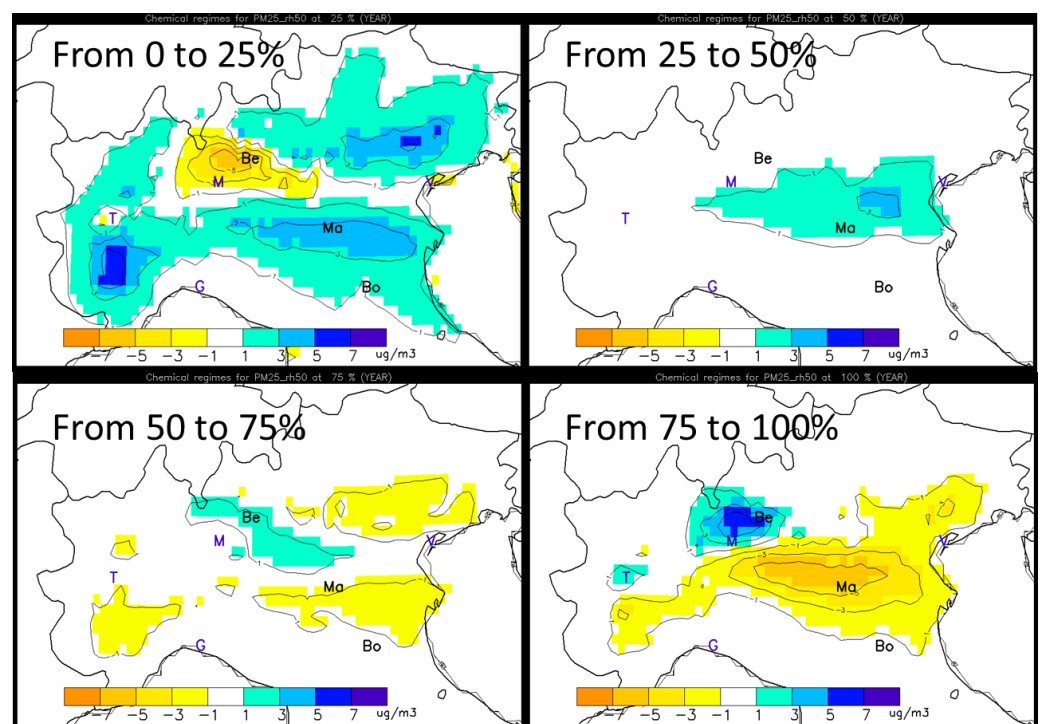

**Figure 5: Yearly averaged chemical regimes obtained for a 25% emission reduction starting at different levels of emissions corresponding to α=0, 25, 50 and 75. The maps represent the $(P_{NO_x} - P_{NH_3})$ between the starting and ending levels (unit: µg/m3) showing the NOx-and NH3 sensitive areas in blue and yellow, respectively. The symbols Be, Ma and Bo indicate the location of Bergamo, Mantova and Bologna, respectively. Other cities are indicated by their first letter for convenience: Venice, Milan, Turin and Genova.**

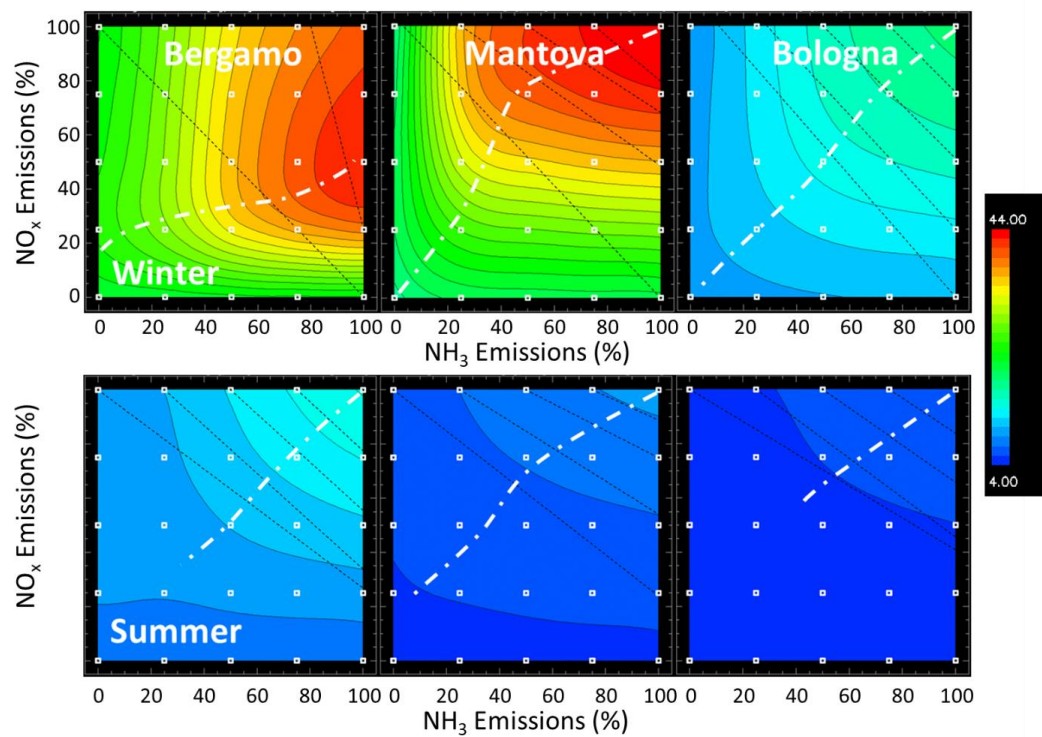

**Figure 6: PM2.5 isopleths during Winter (top row) and Summer (bottom row) at the three locations of interest (see maps). PM2.5 concentrations are expressed in as a function of the intensity of the NOx (Y-axis) and NH3 (X-axis) emissions, respectively. The overlaid dashed oblique lines on each diagram connect similar PM2.5 concentration values for single NOx and NH3 reductions. The more vertical are these lines, the larger is the NH3 abatement impact compared to the NOx abatement impact; The more horizontal they are, the larger is the NOx abatement impact compared to the NH3 abatement impact. The dashed line delineates the ridge between the NH3 and NOx sensitive areas.**

700



**Figure 7: Wintertime isopleths in Bergamo for the species: dry PM2.5, O3, NO2, sulfate (SO4), nitrate (NO3), ammonium (NH4), organic matter (OM), anthropogenic and biogenic secondary aerosols (ASOA and BSOA, respectively). The two numbers on the vertical axis indicate concentration values for the base case and at the 50% NOx emission level.**


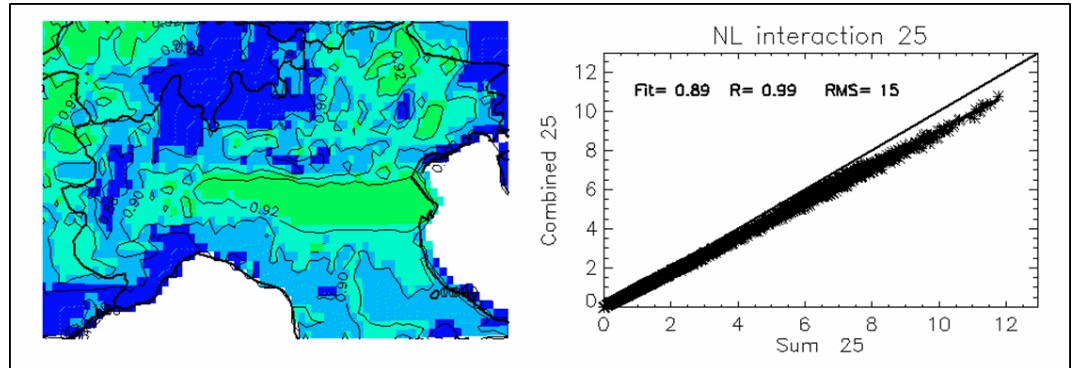

Figure 8: Yearly mean non-linear interaction term for a 25% reduction in NOx and NH3. The map (left) shows the ratio $\frac{P_{NO_xNH_3}^{25\%}}{P_{NO_x}^{25\%}+P_{NH_3}^{25\%}}$ whereas the scatter plot compares the sum of the impacts of the two-single reductions (X-axis) with the impact of the combined emission reduction (µg/m3) (Y-axis).

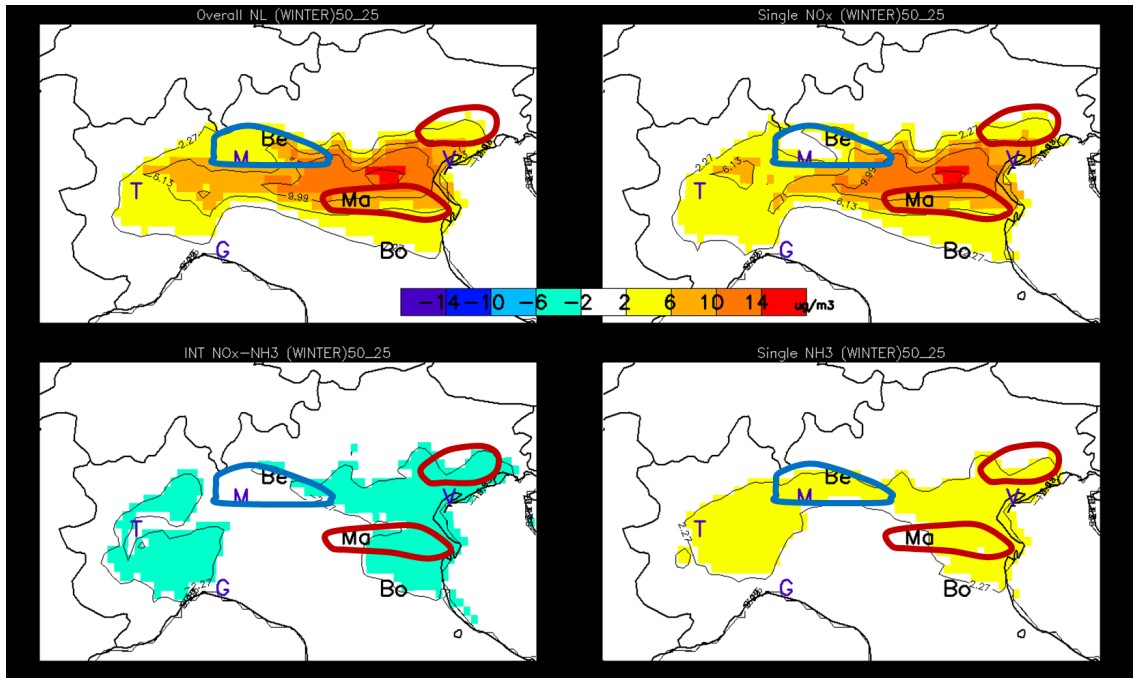

Figure 9: Wintertime maps of the overall non-linearity term (top left) and of its components: the single NOx non-linearity term (SNOx – top right), the single NH3 non-linearity term (SNH3 – bottom right) and the NOx-NH3 interaction term (INOXNH3 – bottom left). The three locations of interest: Bergamo, Mantova and Bologna are indicated by their first two letters while other cities are indicated with their first letter for convenience (Venice, Milan, Turin and Genova). The hand drawn contours roughly indicate the central position of the NH3 (blue) and NOx (red) sensitive regime areas (see Figure 3 - right).



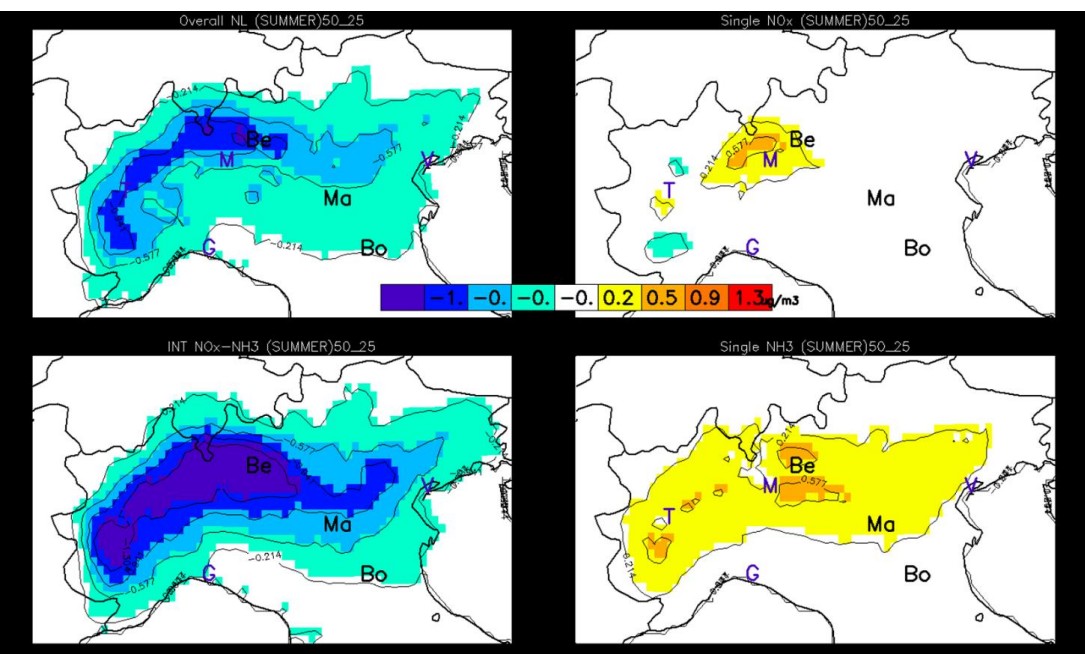

**Figure 10: same as Figure 9 but for summer time.**



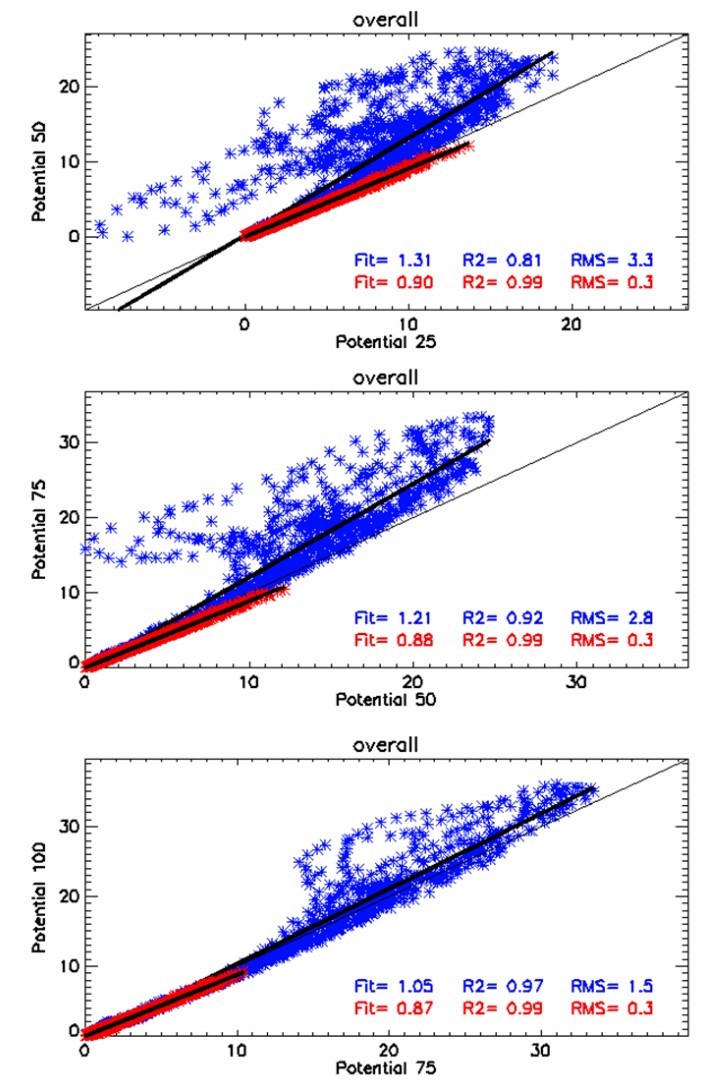

**Figure 11: Changes of the overall non-linearity terms from 25 to 50% (top), from 50 to 75% (middle) and from 75 to 100% (bottom) in NOx and NH3. Each point represents one land grid cell within the domain for wintertime (blue) and summer time (red). The "fit" parameter indicates the slope of the regression line while R2 and RMS provide information on the coefficient of determination and the root mean square error, respectively.**



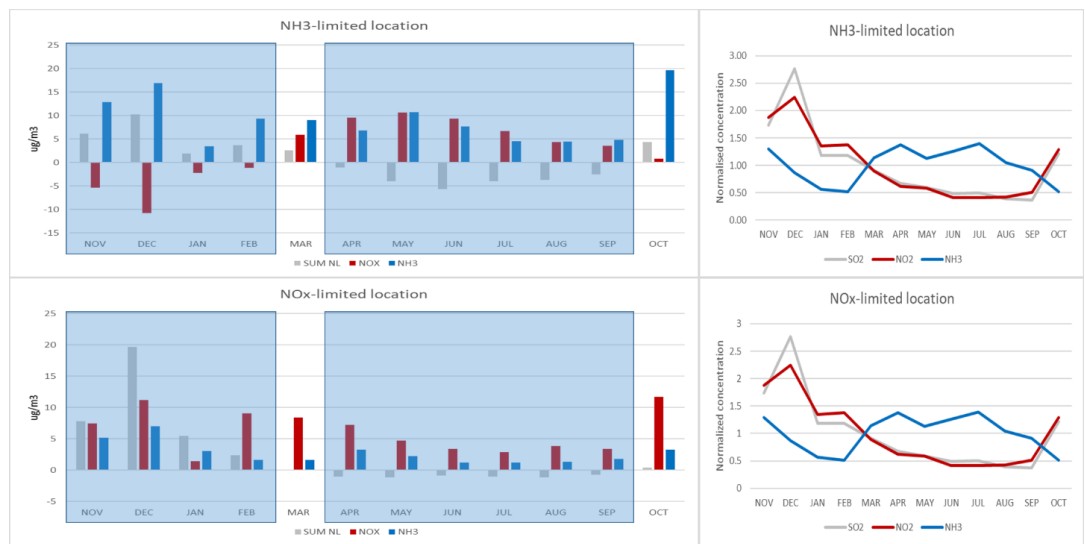

**Figure 12: Monthly averaged responses to NH3 (blue), NOx (red) reductions (25%) and interaction terms (grey) at two locations: Bergamo (top) and Mantova (bottom). The right panel shows the monthly evolution of the concentrations of NO2, NH3 and SO2 at those two locations. Note that concentrations are normalised by their average values.**





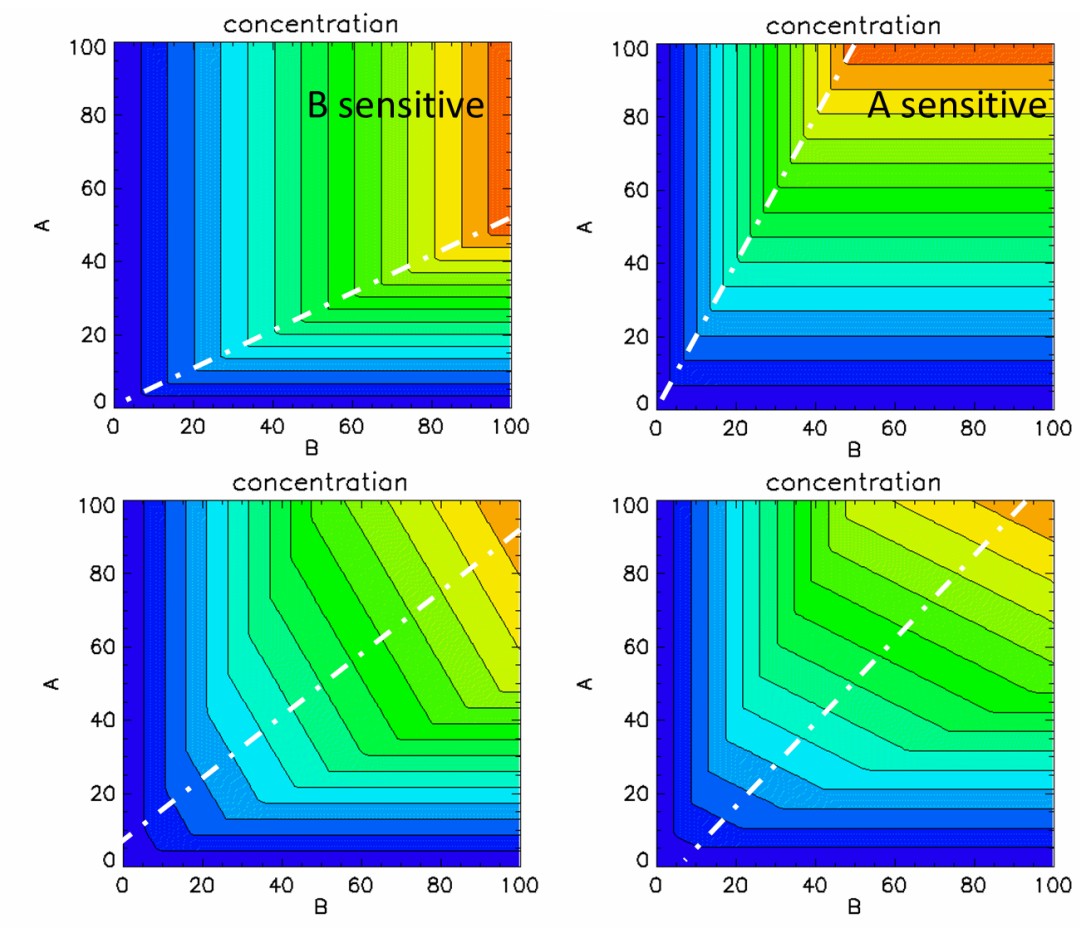

**Figure 13: Isopleths for a simple theoretical system consisting of two emission precursors (A and B) competing through non-linear reactions to the concentration of a pollutant. See details in the text.**
