# Peer review of "Non-linear response of PM2.5 to changes in NOx and NH3 emissions in the Po basin (Italy): consequences for air quality plans"

_Atmospheric Chemistry and Physics, 2021_

## Author Response (AR1)

**Reviewer 1**

**General comments**

The paper "Non-linear response of PM2.5 to changes in NOx and NH3 emissions in the Po basin (Italy): consequences for air quality plans" by Thunis et al. deals with a key aspect of air quality modelling that is the support in developing air quality plans.

The paper provides an interesting discussion about the role of non-linear atmospheric processes and their possible consequences on the robustness of the information that can be derived to support policy makers.

The paper takes advantage of previous works that defined a methodological framework as well as a set of indicators that allow to identify and quantify the role of main non-linear process from both a spatial and temporal point of view.

Therefore, the paper fits the scope of ACP.

The paper is well written, with concise and clear statements, and it does not require any substantial review of syntax and language.

We thank the reviewer for the positive appreciation of the paper and for his/her useful and constructive comments that helped improving our manuscript.

However, before publication, there are a few issues that should be addressed by the authors, that are detailed in the following:

- The discussion on the results shown in a few key figures of the paper requires some checks and additional details concerning both the presentation of the results, but also the related discussion (see Specific comments for additional details)

  Each of the Reviewer's specific comments has been addressed (see detailed responses below for each of these comments)

- The link between "domain-based" emission reductions and the analysis at specific receptors should be partially revised and integrated (see Specific comments for additional details)

  These specific comments have been addressed. We added the following sentence at the end of Section 6.

  "We must however remember that these isopleths derive from uniform emission reduction over the whole domain. When comparing sites providing different answers to the same "domain-based" policy, it remains challenging to define a single domain-based policy reducing PM in both sites."

- Authors presented their results using several and very illustrative type of plots (maps, scatter diagrams, isopleths), but it seems that, in some case, results presents some inconsistencies; moreover, authors could consider to maybe introduce some additional links between the different kind of figure to support the discussion (e.g. between isopleths and maps,…). Annex 2 may deserve a partial review.

  We have addressed and resolved the inconsistencies noted by the Reviewer in his specific comments. We also added links between figures where possible and reviewed annex 2.

There are also a few minor mistakes in references to figures to be fixed and some minor improvements in figures and captions that could be introduced (see Specific comments for additional details).

These mistakes and improvements have been implemented as required in specific comments.

**Specific comments and Technical corrections**

P5  R207 – Are two empty brackets to be removed?

Yes indeed. They have been removed.

P6 R221 – Being "G" a ratio, the linear legend is probably not the most appropriate because it does not allow a "symmetric" comparison of value lower greater and lower than 1 (e.g. G = 2.0 and G=0.5)

In fact the comparison of the G-ratio and sensitivity approaches indicate a threshold value for the G ratio around 6 or 7. A non-linear legend (e.g. logarithmic) would tend to emphasize too much the low-values range. We therefore kept the linear legend but modified the scale so that values below 1 have now a unique color code (in other words, we split the existing "0-2" range so that the 0-1 range explicitly appears to facilitate the comparison).

P7 R294 – Figure 7? If yes, the content of this figure should be better introduced, because it is then used for discussion

It is indeed referring to Figure 7. We re-arranged the paragraph and added some numbers in the text to strengthen the links to the figure. The paragraph now reads as:

The impact of NOx emission reductions on the concentration of various pollutants in Bergamo during wintertime is illustrated in Figure 7. As expected, a 50% reduction in $NO_x$ emissions leads to a decrease in $NO_2$ concentration (from 47 to 28 µg m$^{-3}$, i.e. a factor of 1.7). In contrast, $O_3$ concentration increases from 8 to 16 µg m$^{-3}$, roughly a factor of two. These compensating changes result in a small increase in $NO_3$ radical production (Eq. 9), the initial step of the major pathway of wintertime $HNO_3$ and nitrate formation (Kenagy et al., 2018).

P9 R354-356 – Authors should comment why interaction term is negative, although this result is expected because it is well known that a reduction of only NOX implies a reduction of both NO3- and NH4+ and the same happens when reducing only NH3, therefore a simultaneous reduction of both precursors is lower than the sum of the two.

We commented on the negativity of the interaction. The updated version now reads as:

This negativity can be explained by the fact that a reduction of only $NO_x$ implies a reduction of both $NO_3^-$ and $NH_4^+$ and the same happens when reducing only NH3, therefore a simultaneous reduction of both precursors is lower than the sum of the two.).

P9 R363 – May authors better explain how the "hyperbolic shapes indicate this negative non-linearity in the NH3-NOx interaction"?

We rephrased the entire paragraph (L386 – 392, revised version). In addition to the lines above, the paragraph continues as follows:

Single impacts would therefore lead to an overestimation (of about 10%) in PM2.5 reduction if added up to extrapolate linearly the impact of combined 25% NOx and NH3 emission reductions on yearly averaged PM2.5 concentrations. This result is expected for what concerns particulate NH4NO3, as a consequence of the gas/particle equilibrium described in Eq. 5, although non-linear relationships between NOx emissions and HNO3 concentrations also play a role. Qualitatively, this negative interaction is also highlighted by the hyperbolic shapes of the PM2.5 isopleths determined for 3 different sites of the domain (Figure 6). As discussed in Annex 2, linearity would result in isopleths parallel to the descending diagonal lines.

P9 R379 – Figure 7?

Right. It has been corrected.

P9 R373-379 – Figure 9 may deserve additional discussions… a few suggestions:

1. Top right panel states that is clearly higher than in several areas and, moreover, looking for example at Bergamo in Figure 7, it is not even clear if there is a change in sign (i.e. if they indicate a PM reduction or increase, due to NOx reduction). Does this imply that the extrapolation of scenario results for NOX, in terms of policy, is particularly critical in such emission reduction range?

   Unfortunately, we did not understand the first question. Regarding the final question, it might indeed be critical to extrapolate scenario results east of Bergamo, where the wintertime non-linearities are quite important. At Bergamo, non-linearities are however quite moderate despite the negative response of PM to NOx reductions. We tried to better distinguish these two processes (negative response and non-linearity) in the text. The document was updated as follows:

   Putting all the arguments together, it follows that wintertime ammonium nitrate formation over Bergamo is most probably controlled by $NO_3$ radical formation (9). The fact that this behaviour is observed in Bergamo and not in Mantova or Bologna is due to the much larger $NO_2$ levels in the Bergamo - Milano area (above 50 µg m$^{-3}$ during winter, Fig.2). Such large $NO_2$ levels are also simulated locally over the Turin area, and also lead to a slightly $NH_3$ sensitive regime there despite a G-ratio well above unity. Beyond 50% $NO_x$ reduction, $NH_4NO_3$ formation decreases because $NO_2$ decreases more rapidly than ozone increases up to its maximum (at 75% $NO_x$ emission reduction, see Figure 7).

   The negative response of NH4NO3 to NOx emissions explains the apparent discrepancies with the analysis of G-ratio indicating stronger sensitivity to $NO_3^-(p)$. Simply, sensitivity to $NO_3^-(p)$ concentration cannot be extrapolated to sensitivity to NOx emissions in case of the above shown negative response. Total nitrate is less abundant than free ammonia (defined as $NH_3(g) + NH_4^+(p) - 2SO_4^{2-}(p)$), but NOx emission reductions below about 50% do not reduce it, and NH3 emission reductions are thus more efficient. In this respect, the G-ratio does not provide information about negative responses

2. In other works, same authors state that below 50% reduction the impact approach (i.e. emission reduction approach) can be considered linear up to 50% reduction: does this result allow to confirm such assumption?

   We added this sentence in the document to discuss this point (L418-424 in revised paper):

   In a previous work, Thunis et al. (2015) quantified the non-linearity of model responses to emission reductions in three areas in Europe, among which the Po Basin. In each area, three locations (urban, suburban and rural) were selected to analyse the non-linearities associated to different temporal averages. One of the author's conclusions was that non-linearities remain relatively low for yearly averaged responses. Although the results presented here show important nonlinearities, these occur mainly during wintertime and are limited to specific areas. It is also worth noting that these non-linear behaviours (for moderate emission reductions up to -50%) only occur in the Po Basin Clappier et al. (2021).

3. Bottom right panel states that is slightly higher than  in some areas: any explanation?

   We unfortunately did not understand this question. We however added the following sentence to explain the bottom right panel:

   "Similarly to NOx, the NH3 single non-linearities (Figure 9, bottom right) are positive but weaker, indicating slightly larger potential impacts for emission reductions in the range 25-50% than in the 0-25% range"

4. Is the comment concerning bottom left panel correct? Should it indicate that the interaction term at 50% is lower (i.e. "more negative") than the corresponding term at 25%, isn't it?

Right. We corrected the text as follows:

"Finally, the non-linear interactions (Figure 9, bottom left) are mostly negative, indicating that the interaction term at 50% is lower (i.e. "more negative") than the corresponding term at 25%, pointing out to a strengthening of the non-linearities when more intense emission reductions are considered.

P10 R390 – Again… bottom left panel in figure 10, in principle, should indicate that interaction term at 50% is lower than the corresponding term at 25%, not providing information on the sign of each (P50 and P25) interaction term. Then, knowing that P25 is lower than 0, we can conclude that also P50 is also negative

Right. We corrected the sentence as follows:

"Finally, the $NO_x$-$NH_3$ non-linear interaction terms (Figure 9, bottom left) are mostly negative, indicating that the interaction term for -50% emission reductions is more negative than the corresponding term for -25%, pointing out to a strengthening of the $NO_x$-$NH_3$ non-linearity when more intense emission reductions are considered)".

P10 R392-394 – Maybe it could be more readable putting this comment after having introduced figure 11

We removed this comment, as it was redundant with the information of the next paragraph.

P10 R395 – Does figure 11 show the last four terms of term of equation 4?

If yes, it is not clear to me the range of plotted values. For example, in summer case, figure 10 (Top left) states that the last three terms in the above equation ranges between -1. and 0. and figure 8 states that the first term of the above equation is negative; so, why in figure 11 (top panel) do red values range between 0 and 15 ug/m3?

The axis of this figure were indeed confusing. The X- and Y-axis represent the overall potentials at two levels of emission reductions. This is why the numbers reach 15 ug/m3 on the axis. The difference between the two (i.e. the difference with the 1:1 line in the scatter) represents the overall non-linearity (sum of the 4 terms in equation 4). We have clarified the text in the document, clarified the figure and added the following sentence in the figure caption:

"Changes of the overall non-linearity terms from 25 to 50% (top), from 50 to 75% (middle) and from 75 to 100% (bottom) in NOx and NH3. The overall non-linearity is visualised as the distance from the 1:1 diagonal, i.e. the difference between the overall potential at two levels of emission reduction, X- and Y-axis."

P10 – R426-430 – Isopleth diagrams represent a very interesting tool to assess the impact of different NOx and NH3 emission reductions on PM2.5 concentration, but authors should always point out that all the results, even if they are analysed at a single cell, derive from uniform emission reduction over the whole domain; therefore the key aspect to be highlighted is that, when comparing sites providing different answers to the same "domain-based" policy, it is not trivial to define a single domain-based policy reducing PM in both sites either identifying different policies (for example for specific sub-regions) again reducing PM in both sites.

Right. We modified the last bullet as follows:

Although they are location specific, PM2.5 isopleths represent an interesting tool to assess the impact of different NOx and NH3 emission reductions on PM2.5 concentration. They indeed allow visualising in one single diagram the impact of any type of reductions on concentrations in a single grid cell (or set of grid cells). We must however remember that these isopleths derive from uniform emission reduction over the whole domain. Comparing sites where PM2.5 responses to the same "domain-wide" policy are different, it appears challenging to define a single domain-wide policy efficiently reducing PM at all locations.

P11 – R457-458 – Figure 9 seems stating that the system is rather sensitive to emission reduction between 25 and 50%, at least for NOx, so do authors think that their statement "they are quite insensitive to the emission reduction levels at least up to -50% " is totally correct?

We agree with the reviewer. As indicated in our responses above, this statement was indeed not totally correct. We now rephrased this sentence as:

While PM2.5 chemical regimes, determined by the relative importance of the NOx vs. NH3 responses to emission reductions, show large variations seasonally and spatially, they are not very sensitive to moderate (up to 50-60%) emission reductions. Beyond 25% emission reduction strength, responses of PM2.5 concentrations to NOx emission reductions become non-linear in certain areas of the Po basin mainly during wintertime.

P12 R493 – Clappier et al is 2020 or 2021?

2021. The date has been changed to 2021 at relevant places in the document

P16 R636 – "March" instead of "April"?

Indeed, changed.

P16 R654-656 – Is the relation between sensitivity and triangles to be reversed?

Indeed, changed.

P16 R657-662 – Explanation in second bullet is not very clear. Please check and rephrase it

We have re-written this paragraph that was indeed confusing. It now reads as:

"In the case of a single A-sensitive day (top right) with EB(100)=2EA(100), the concentration of compound C remains unchanged for emission reductions of B up to 50% while its concentration reacts in a linear way to emission changes of A from 0 to 100%. Between the base case and a reduction level of 50%, the A-gradient is therefore larger than the B-gradient. This implies that the B-gradient is larger than the A-gradient between the 50 and 100% reduction levels because we know that for a 100% reduction of A or B, the concentration must be zero. In our simple example, the gradient of B is zero from 0 to 50% but is twice as large as A between 50 to 100%."

P17 R670 – Caption should indicate that SIA are expressed in percentage

Changed as suggested.

P19 R691 – A blank is missing between "to" and "alpha"

Corrected.

P22 R710 – A legend for colours is missing in figure 8 ( left). Does figure 8 (right) shows daily mean values of each grid cell?

We removed the left figure as it was showing rather confusing patterns due to the very low values plotted. We clarified the explanations in the caption (now: each point represents a yearly average for one grid cell in the domain) and improved the figure.

P23 Figure10 – Which is the lowest value in the colour legend?

The colour scale has been corrected.

**Reviewer 2**

In this work, Thunis et al. simulate and analyze sensitivities of $PM_{2.5}$ to $NO_x$ and $NH_3$ emissions over the Po valley in order to provide guidance for the designing of future air quality plans in the area. This study is of definite interest to the ACP audience since the Po basin is one of the most polluted regions in Europe and, therefore, it is of high importance to elucidate the complex chemical processes that lead to frequent exceedances of the EU $PM_{2.5}$ concentration limits. The manuscript is well written, the methodology is scientifically sound, and the discussion is clear. However, the authors need to provide more details regarding the underlying aerosol (both organic and inorganic) formation processes considered by their model which is a prerequisite for the thorough interpretation of the results presented in this study. Furthermore, the illustration of the results needs to be improved. Overall, I recommend this study for publication. Below are a few comments to be considered prior to publication.

We would like to thank the reviewer for the positive appreciation of the paper and his/her useful and constructive comments that helped improving our manuscript.

**General comments:**

1. The authors should include a description of the modelling framework used for the present study. The reader should be aware of the mechanistic details of the model in order to validate and interpret the simulated responses presented here. The model description should include the gas phase chemistry scheme used (including, if applicable, the heterogeneous chemistry), the thermodynamic aerosol model used for the formation of SIA, and the organic aerosol framework used for the formation of SOA (including the main NMVOCs used as SOA precursors and their photochemistry).

We agree with the Reviewer and added paragraphs (L107 – 131, new version) to address these aspects in the methodological section.

The gas-phase chemistry is based on the evolution of the so-called "EMEP scheme" as described in Simpson et al. (2012) and references therein. The chemical scheme couples the sulfur and nitrogen chemistry to the photochemistry using about 140 reactions between 70 species (Andersson-Sköld and Simpson, 1999; Simpson et al. 2012). In the EMEP Status Report 1/2004 (Fagerli et al., 2004) the reactions are described that cover acidification, eutrophication and ammonium chemistry. The aqueous phase chemistry describes the formation of sulfate in clouds via $SO_2$ oxidation by ozone and $H_2O_2$ and catalysed by metal ions. An important pathway of particulate nitrate formation is through the hydrolysis of $N_2O_5$ on wet aerosol surfaces that converts $NO_x$ into $HNO_3$. More information on the chemical equations is given in Simpson et al. (2012), section 7.

The EMEP model has two size fractions for aerosols, fine aerosol ($PM_{2.5}$) and coarse aerosol ($PM_{10-2.5}$). The aerosol components presently accounted for are sulfate ($SO_4^{2-}$), nitrate ($NO_3^-$), ammonium ($NH_4^+$), anthropogenic primary PM and sea salt.

For inorganic aerosols, EMEP uses the MARS equilibrium module to calculate the partitioning between gas and fine-mode aerosol phase in the system of $SO_4^{2-}$, HNO3, $NO_3^-$, NH3 and $NH_4^+$ (Binkowski and Shankar, 1995). Aerosol water is calculated to account for particle water within the PM2.5 mass, which depends on the mass of soluble PM fraction and on the type of salt mixture in particles. Sea salt (sodium chloride) and dust components are not accounted for by MARS, which might lead to PM underestimations close to coastal sites and where the dust contribution is important. More information on the gas and aerosol partitioning is given in Simpson et al. (2012), section 7.6.

Regarding secondary organic aerosols (SOA), the EmChem09soa scheme is used, which is a simplified version of the so-called volatility basis set (VBS) approach (Robinson et al., 2007; Donahue et al., 2009). The VBS mechanism is discussed in detail in Bergström et al. (2012). The main differences between the VBS schemes and EmChem09soa is that all primary organic aerosol (POA) emissions are treated as non-volatile in EmChem09soa. This is done to keep the emission totals of both PM and VOC components the same as in the official emission inventories. The semi-volatile biogenic and anthropogenic SOA species are assumed to further oxidise (also known as ageing process) in the atmosphere by OH-reactions. This will lead to a reduction in volatility for the SOA, and thereby increased partitioning to the particle phase. More information on SOA is given in Simpson et al. (2012), section 7.7.

2. It has been recently suggested that aerosol pH and liquid water content can be used to determine when PM is sensitive to NH3 and/or HNO3 availability (Nenes et al., 2020). It could be useful to directly compare (or at least discuss) the results of this method against the indicators presented here

We think that a direct comparison with this approach is difficult (because we did not directly analyse modelled water content and pH in this work) and beyond the scope of this paper, which is to determine the sensitivity of PM2.5 concentrations to reductions in NH3 and NOx emissions. We chose to compare our results based on emission reduction scenarios to the information provided by the G-ratio because the variables used in both approaches were the same.

We also believe that considering the aerosol pH and water content as indicators for PM sensitivity to NH3 and/or HNO3 availability would be more of a static approach, therefore quite close to the G-ratio concept. Indeed, it looks at the status of the aerosol, which (as seen from the G-ratio) cannot tell about changes occurring when relatively large emission reductions are applied

**Specific comments:**

1. Page 1, line 21 (and hereafter): Place 2.5 as subscript in PM2.5 (i.e., PM$_5$).

   Done as suggested.

2. Page 2, line 55: Please rephrase. By definition, NMVOCs are not low-volatility.

   We rephrased as: "a vast range of NMVOCs"

3. Page 2, lines 58-76: Important studies regarding the effectiveness of NH$_3$ reductions to control PM concentrations are Pozzer et al. (2017);Guo et al. (2018);Nenes et al. (2020).

   Thanks for the suggested references, they have been added.

4. Page 2, line 81: You can also add Tsimpidi et al. (2016) for a detailed comparison of OA composition against AMS-measurements which elucidates important OA formation pathways that are still missing by the models

   Thanks for the suggested reference, it has been added.

5. Section 2: The methodology should include a model description. It is of prime importance to briefly present the details of the model such as the thermodynamic model used. A brief description of the organic aerosol framework is also important since the NO$_x$ reduction can impact the oxidant levels and thus the SOA formation.

   We added several paragraphs in the methodological section to address this point (see general comment - point 1)

6. Pages 4, lines 170-176: Furthermore, low temperatures during winter favor the aerosol phase during the equilibrium partitioning of semi-volatile components (e.g. ammonium nitrate).

   We added the sentence proposed by the Reviewer in the text.

   The increased emissions from the residential sector (heating, especially wood burning) foster this process (Ricciardelli et al., 2017; Hakimzadeh et al., 2020). Wintertime low temperatures also favour the partitioning of semi-volatile components (e.g. ammonium nitrate) towards the particulate phase. Overall, the relative contribution of secondary inorganic particles (SIA) ranges between 40 and 50%, regardless of the season and is quite homogeneously distributed spatially over the entire area.

7. Page 5, equations 5, 6: These belong to model description. Does the model include any heterogeneous chemistry?

   Yes, it does. This aspect is now discussed in the methodological section (see general comment - point 1)

8. Page 5 line 208: Correct "of" with "or"

   Done as suggested.

9. Page 8, equations 10-15: Are these included in the model?

   Yes, these equations are included in the model. This aspect is now discussed in the methodological section (see general comment - point 1).

10. Page 8, lines 327-335: Maybe it is worth drawing a map of the VOC/NOx concentration ratio to illustrate the NOx-limited (i.e., with ratio values higher than 5.5) and NOx-saturated areas of the model domain (Tsimpidi et al., 2008).

    Saying that the chemical regime is VOC limited is in fact not enough. For a given NOx reduction, we also need that the O3 increase be stronger than the related NO2 decrease (because of equation NO2 + O3 -> NO3). This is certainly the case for Bergamo with a 100 % increase of O3 for -25% NOx emissions. These changes are more related to titration, so simply the NOx concentration is a good tracer, or NO2 which we have already and which shows a maximum in both Milano, Bergamo region. We therefore believe that a VOC/NOx map would not help in this context

11. Page 8, line 330: correct "that" with "than"

    Done.

12. Page 8, line 333: Does the model include heterogeneous or aqueous phase production of SOA as implied here?

    There are NO aqueous phase reactions of SOA. We mentioned this in the description of the methodology by adding the following paragraph:

    Regarding secondary organic aerosols (SOA), the EmChem09soa scheme is used, which is a simplified version of the so-called volatility basis set (VBS) approach (Robinson et al., 2007; Donahue et al., 2009). The VBS mechanism is discussed in detail in Bergström et al. (2012). The main differences between the VBS schemes and EmChem09soa is that all primary organic aerosol (POA) emissions are treated as non-volatile in EmChem09soa. This is done to keep the emission totals of both PM and VOC components the same as in the official emission inventories. The semi-volatile biogenic and anthropogenic SOA species are assumed to further oxidise (also known as ageing process) in the atmosphere by OH-reactions. This will lead to a reduction in volatility for the SOA, and thereby increased partitioning to the particle phase. More information on SOA is given in Simpson et al. (2012), section 7.7.

13. Page 17, Figure 1 (and hereafter): In order to improve the illustration of the plotted maps I suggest to increase the number of colors used, draw the border lines bolder, and move the color bars to the right of the figure (using only one instead of three).

    Done as suggested.

14. Pages 18-19, Figures 3, 5 (and thereafter): I suggest using diverging colors for the figures (e.g., shades of blue colors for negatives and shades of red colors for positives)

    Done as suggested.

15. Page 18, Figure 4: I suggest using diverging colors below and over the value of 1 for the G-ratio.

    We agree that 1 would be the logic value to use as separator. However, we show that the G-ratio map is comparable to the regime map only when this separator is around 6. This is why we use this value to separate the low and high G-ratio areas. By the way, we changes the colour scale so that values within the "0-1" range appear explicitly.

16. Page 20, Figure 6: Values are missing on the y-axis for Summer. The color-bar title is also missing. Please add more value points in the color-bar as well.

    Figure has been updated as suggested.

17. Page 21, Figure 7: The color-bar is missing

    Color bars have been added.

18. Page 22, Figure 8: Please consider revising the map on the left. The way it is represented is not clear.

    The map is indeed challenging because all values are small. As the scatter plots deliver this information already, the map has been removed.

19. Page 26, Figure 13: Color-bar is missing

    The color bar has not been added as precise values do not matter in this example but an explanation of the colors used has been added in the caption.

References:

Guo, H., Otjes, R., Schlag, P., Kiendler-Scharr, A., Nenes, A., and Weber, R. J.: Effectiveness of ammonia reduction on control of fine particle nitrate, Atmos. Chem. Phys., 18, 12241-12256, 10.5194/acp-18-12241-2018, 2018.

Nenes, A., Pandis, S. N., Weber, R. J., and Russell, A.: Aerosol pH and liquid water content determine when particulate matter is sensitive to ammonia and nitrate availability, Atmos. Chem. Phys., 20, 3249-3258, 10.5194/acp-20-3249-2020, 2020.

Pozzer, A., Tsimpidi, A. P., Karydis, V. A., de Meij, A., and Lelieveld, J.: Impact of agricultural emission reductions on fine-particulate matter and public health, Atmospheric Chemistry and Physics, 17, 12813-12826, 10.5194/acp-17-12813-2017, 2017.

Tsimpidi, A. P., Karydis, V. A., and Pandis, S. N.: Response of Fine Particulate Matter to Emission Changes of Oxides of Nitrogen and-Anthropogenic Volatile Organic Compounds in the Eastern United States, J. Air Waste Manage. Assoc., 58, 1463-1473, 10.3155/1047-3289.58.11.1463, 2008.

Tsimpidi, A. P., Karydis, V. A., Pandis, S. N., and Lelieveld, J.: Global combustion sources of organic aerosols: model comparison with 84 AMS factor-analysis data sets, Atmos. Chem. Phys., 16, 8939-8962, 10.5194/acp-16-8939-2016, 2016.